# Fluorescent Probes with Förster Resonance Energy Transfer Function for Monitoring the Gelation and Formation of Nanoparticles Based on Chitosan Copolymers

**DOI:** 10.3390/jfb14080401

**Published:** 2023-07-27

**Authors:** Igor D. Zlotnikov, Ivan V. Savchenko, Elena V. Kudryashova

**Affiliations:** Faculty of Chemistry, Lomonosov Moscow State University, Leninskie Gory, 1/3, 119991 Moscow, Russia; zlotnikovid@my.msu.ru (I.D.Z.);

**Keywords:** FRET, chitosan, nanoparticles, gelation, fluorescent probe, ovalbumin

## Abstract

Nanogel-forming polymers such as chitosan and alginic acid have a number of practical applications in the fields of drug delivery, food technology and agrotechnology as biocompatible, biodegradable polymers. Unlike bulk macrogel formation, which is followed by visually or easily detectable changes and physical parameters, such as viscosity or turbidity, the formation of nanogels is not followed by such changes and is therefore very difficult to track. The counterflow extrusion method (or analogues) enables gel nanoparticle formation for certain polymers, including chitosan and its derivatives. DLS or TEM, which are typically used for their characterization, only allow for the study of the already-formed nanoparticles. Alternatively, one might introduce a fluorescent dye into the gel-forming polymer, with the purpose of monitoring the effect of its microenvironment on the fluorescence spectra. But apparently, this approach does not provide a sufficiently specific signal, as the microenvironment may be affected by a big number of various factors (such as pH changes) including but not limited to gel formation per se. Here, we propose a new approach, based on the FRET effect, which we believe is much more specific and enables the elucidation of nanogel formation process in real time. Tryptophan-Pyrene is suggested as one of the donor–acceptor pairs, yielding the FRET effect when the two compounds are in close proximity to one another. We covalently attached Pyrene (the acceptor) to the chitosan (or PEG-chitosan) polymeric chain. The amount of introduced Pyrene was low enough to produce no significant effect on the properties of the resulting gel nanoparticles, but high enough to detect the FRET effect upon its interaction with Trp. When the Pyr-modified chitosan and Trp are both present in the solution, no FRET effect is observed. But as soon as the gel formation is initiated using the counterflow extrusion method, the FRET effect is easily detectable, manifested in a sharp increase in the fluorescence intensity of the pyrene acceptor and reflecting the gel formation process in real time. Apparently, the gel formation promotes the Trp-Pyr stacking interaction, which is deemed necessary for the FRET effect, and which does not occur in the solution. Further, we observed a similar FRET effect when the chitosan gel formation is a result of the covalent crosslinking of chitosan chains with genipin. Interestingly, using ovalbumin, having numerous Trp exposed on the protein surface instead of individual Trp yields a FRET effect similar to Trp. In all cases, we were able to detect the pH-, concentration- and temperature-dependent behaviors of the polymers as well as the kinetics of the gel formation for both nanogels and macrogels. These findings indicate a broad applicability of FRET-based analysis in biomedical practice, ranging from the optimization of gel formation to the encapsulation of therapeutic agents to food and biomedical technologies.

## 1. Introduction

Chitosan (Chit) is a linear copolymer of D-glucosamine and N-acetyl-D-glucosamine bonded together by a β(1–4) bond, usually obtained from natural chitin polysaccharide by partial or complete deacetylation. Due to its properties, including cellular and tissue biocompatibility as well as biodegradability, chitosan has a wide range of applications and is now being implemented in many areas [1] including environmental protection, medicine, food production, drug production, etc. This polymer is a promising structural material in medicine because of its antimicrobial [2], antioxidant and regenerative activity [3], as well as the possibility of the effective encapsulation of drugs, proteins [4], nucleic acids [5], eukaryotic cells [6], bacteria and viruses [7,8]. 

One of the features of chitosan is its ability to form nano- and microgels, the properties of which can be changed in a wide range due to the chemical modification of the functional groups of the polymer, its crosslinking with other macromolecules as well as its non-covalent interaction with proteins and DNA [9,10]. Specific methods for obtaining chitosan nanoparticles of various shapes and molecular structure have been developed since their first production and include ionotropic gelation methods [11], SCF technology, microemulsions, the reverse micellar method [12] and electrospinning [13].

It has been shown that chitosan is able to form gels in the presence of crosslinking agents, among which succinic anhydride [14,15,16,17], glutaraldehyde [18,19] and genipin are most often used, which covalently bind to several chitosan polymer molecules and act as bridges [20,21]. Genipin is an aglycone derived from an iridoid glycoside from the fruits of jasmine gardenia, and it is 10,000 times less toxic than glutaraldehyde. Moreover, genipin has anti-inflammatory and anti-angiogenic properties, is used for liver disorders, and stimulates the release of insulin. Crosslinking with other macromolecules and polymers is often used to change the physicochemical parameters of gels—mechanical strength, sensitivity to medium acidity, biodegradability, etc. The crosslinking of polymer chains with the simultaneous capture of the fluorophore inside the polymer tangle (chemical gelation) significantly affects the properties of the probe more strongly than physical gelation.

The formation of gels and nanoparticles depends very much on the polymer structure, concentration of the crosslinking agent, the acidity of the medium and the temperature, which can also be used in the development of methods for obtaining particles of a certain size and density [22].

The modification of chitosan by fluorophores is widely used in bio-imaging technology both in fundamental research and in clinical practice [23,24,25]. Upon the polymeric particles, the formation of the fluorophore molecule can be adsorbed by the polymer [26] or bound to it covalently [27]. In the present work, pyrene was used as a covalent probe, and Congo red and malachite green are used as non-covalent fluorophores sensitive to the local microenvironment and the folding of polymer chains.

The physicochemical parameters of polymers prone to gelation, such as solubility, sensitivity to the acidity of the medium, fluorescence intensity and position of the maxima, the optical absorption spectrum, and kinetic and thermodynamic stability, strongly depend on the conditions of the preparation and structural features of the polymer [28,29]. When polymeric nano(micro)particles are formed, the intensity of the dye fluorescence may light up due to changes in the microenvironment or fluorescence quenching during the formation of the particles and due to a significant increase in the viscosity of the solution.

Fluorescent probes can be effectively used to study the formation of particles also from chitosan derivatives, for example, pegylated chitosans (Chit-PEG) that are capable of forming thermogels [30]. Modified chitosan nanoparticles can be applied in medicine due to their ability to include drugs, peptides, nucleic acids, etc., potentially leading to more safe and efficient treatments [9]. Other authors suggest targeted delivery methods based on chitosan derivatives [31,32]. Varying the particle sizes and their physicochemical properties also makes it possible to reduce the side effects of drugs and their effective dose, and at the same time, increase their half-life in blood plasma [19].

When studying the structure of chitosan-based particles, infrared and absorption spectroscopy are often resorted to, which makes it possible to prove the binding of the crosslinking agent with chitosan [33,34].

Despite polyelectrolytes, non-charged polymers are often used as part of drug delivery systems, so they also require a probe. The developed fluorescent probes are sensitive to changes in the hydrophobicity of the microenvironment, for example, during the formation of micelles [35], which was previously demonstrated via the example of a pyrene label [36]. Thus, the three presented fluorescent probes are able to signal the formation of particles from any polymer: Congo red and malachite green are more suitable for polyelectrolytes and pyrene—for micelle-forming polymers.

However, unlike bulk macrogel formation, which is followed visually and by easily detectable changes in physical parameters, such as viscosity or turbidity, the formation of nanogels is not followed by such changes, and is therefore very difficult to track. To detect nanogel particle formation, DLS or TEM are typically used for their characterization, and only allow for the study of the already-formed nanoparticles. Alternatively, one might introduce a fluorescent dye into the gel-forming polymer, with a purpose of monitoring the effect of its microenvironment on the fluorescence spectra. But apparently, this approach does not provide a sufficiently specific signal, as the microenvironment may be affected by a big number of various factors (such as pH changes) including but not limited to gel formation per se.

Förster resonance energy transfer (FRET) can be considered as a promising method of analysis for nanoparticle formation, which consists of the interaction of two fluorophore molecules, resulting in a change in the wavelength of radiation due to the intermediate energy transfer to the acceptor after the donor is excited [37]. Due to the strong dependence of the efficiency of such effect on the chemical environment of fluorophore molecules and the distance between them, FRET is widely used to study the structure of chitosan [38] and materials based on it [39].

Thus, here, we propose a new perspective approach for monitoring nanogel formation together with the drug loading process on the example of nanoparticles based on the chitosan derivatives. The method is based on the FRET effect, which we believe is much more specific and enables the elucidation of the nanogel formation process in real time. Tryptophan-Pyrene is suggested as one of the effective donor–acceptor pairs [40,41], yielding the FRET effect when the two compounds are in close proximity to one another. The inclusion of model drug molecules of tryptophan (serotonin precursor, used for the prevention of mental disorders, amino acid balance) and ovalbumin (vaccines, gels, foams, drug formulation component) in chitosan-based nanogels demonstrates a broad applicability of FRET-based analysis in biomedical practice.

## 2. Materials and Methods

### 2.1. Reagents

Chitosan oligosaccharide lactate 5 kDa (Chit5), activated PEG 5 kDa (N-succinimidyl ester of mono-methoxy poly(ethylene glycol)), ovalbumin, tryptophan (Trp) and 1 M 2,4,6-trinitrobenzenesulfonic acid (TNBS) were purchased from Sigma-Aldrich (St. Louis, MI, USA). Congo red and malachite green were purchased from Reachem, Russia. 1-pyrenebutanoic acid succinimidyl ester was bought from Invitrogen (Molecular Probes, Eugene, OR, USA).

### 2.2. Chitosan PEGylation and Pyrene Modification

Using CD spectroscopy (Jasco J-815 CD Spectrometer, Japan), the degree of deacylation in glycol chitosan samples was determined by the peak at 215 nm corresponding to the absorption of the amide bond, and it was 92–95%. Chit5 was pegylated according to a previously reported technique [30] with a small modification. Chit5-PEG5 copolymer was obtained via reaction between Chit5 amino groups and activated carboxylic group of PEG. The samples of the components in the same mass ratio were mixed and dissolved in PBS, and the mixture was heated to 60 °C and incubated for 4 h. Copolymer was freeze-dried at –70 °C (Edwards 5, West Susse, UK).

Chit5 and Chit5-PEG in concentrations of 1 mg/mL were pyrene-modified via amino group reaction with activated pyrene (1-pyrenebutanoic acid succinimidyl ester) in PBS (pH 7.4) for 72 h at 40 °C. Molar ratio Chit5:pyrene = 2:1, Chit5-PEG:pyrene = 2.5:1.

The degree of Chit5 or Chit5-PEG modification by pyrene was determined using A420 values for amino group adducts with 2,4,6-trinitrobenzenesulfonic acid (TNBS) in 0.02 M sodium borate buffer (pH 9.2). 

### 2.3. Obtaining and Characterizing Nanoparticles

Chit5 and Chit5-PEG nanoparticles were obtained using extrusion (200 or 400 nm membrane, Avanti Polar Lipids) after 1 h incubation of samples (0.01–1 mg/L) at 40 °C. The initiation of gelation was performed by placing a solution with an alkaline medium (pH 8–10), ovalbumin or alginic acid as the counterions in a syringe receiver. Particles’ hydrodynamic diameter sizes and ζ-potentials were measured using Zetasizer Nano S from Malvern (Malvern, UK) (4 mW He–Ne laser, 633 nm, scattering angle 173°) in 0.01 M PBS (pH 7.4). Dynamic light scattering data were analyzed using Zetasizer Software (v. 8.02). Topography, phase and magnitude signal images of the micelles deposited onto freshly cleaved surface of mica were obtained via atomic force microscopy (AFM) using a scanning probe microscope, NTEGRA Prima (NT-MDT, Russia), operated in a semi-contact mode with 15–20 nm peak-to-peak amplitude of the “free air” probe oscillations. Silicon cantilevers NSG01 “Golden” series were used for semi-contact mode (NT-MDT, Russia). Image processing was performed using the Image Analysis software (NT-MDT, Russia).

### 2.4. Fluorescence Approach to Study of Nanoparticles and Gel Formation

Fluorescence emission spectra were recorded on a Varian Cary Eclipse fluorescence spectrometer (Agilent Technologies, Santa Clara, CA, USA). λ_exci_(pyrene) = 330 nm.

### 2.5. FTIR Spectroscopy

FTIR spectra of samples were recorded using a Bruker Tensor27 spectrometer equipped with a liquid-nitrogen-cooled MCT (mercury cadmium telluride) detector, as described earlier [36,42,43,44].

## 3. Results and Discussion

This research is aimed at developing a sensitive approach based on the fluorescent methods (including FRET) to monitor the formation of chitosan-based particles and gels. The main tactic steps used in the work to understand the physicochemical principles of gelation are as follows: (i) the synthesis of polymers and characterization of their molecular composition using FTIR spectroscopy before gelation, as well as obtaining nanogel particles via extrusion with counterion coacervation; (ii) the comparison of the properties of volume-phase macrogel with nanoparticles using fluorescent probes and the optimization of gelation conditions with subsequent particle formation under certain conditions; (iii) the study of the particle’s formation depending on the temperature, pH, polymer concentration, the presence of a crosslinking agent using non-covalent fluorescent probes (malachite green and Congo red) and the determination of gelation conditions and nanoparticle formation; (iiii) the application of a covalently attached pyrene probe to chitosan chain to monitor the nanoparticle formation as well as the encapsulation of the cargo including a small molecule (tryptophan) and a protein (ovalbumin) and the demonstration of a new technique for tracking the drug inclusion using FRET.

### 3.1. Polymer and Particle Characterization

Figure 1a shows the FTIR spectra of Chit5 and Chit5-PEG in D_2_O, which makes it possible to monitor the oscillations of N–H and O–H bonds in the solution. The FTIR spectroscopy provides valuable data regarding the microenvironment and the state of the functional groups. When modifying chitosan amino groups with PEG, an increase in the peak corresponding to the oscillations of the N–H bonds (3400 cm^−1^) is observed in D_2_O as a solvent. The Chit5-PEG conjugate exhibits an intense absorption loss of the C–H bond’s oscillations (weak intensity in simple chitosan) at 2850–2970 cm^−1^, and the intensity of amide 1 and amide 2 peaks, which correspond to the valence oscillations of C=O, and the deformation oscillations of N-H increases. Further confirmation of the presence of both components in the Chit5-PEG conjugate can be drawn from the changes in the shape of the peak corresponding to the oscillations of the C–O bonds (1100 cm^−1^), since the functional groups of both chitosan and PEG contribute to it.

The important parameters characterizing the stability of the colloidal system are the zeta potential and the radius of the particles. Figure 1b,c shows the particle images obtained via atomic force microscopy. Chit5 and Chit5-PEG form particles of 100–350 nm in size, which is consistent with the DLS data (Table 1).

An interesting aspect is the comparison of the volume phase gel (macrogel) and nanogel, as well as the variation of the gel-initiating conditions (pH or counterion). During gel extrusion, particles were formed due to a sharp change in pH from 6 to 10 or due to the interaction with counterion (alginic acid), which is when a counterflow of Chit5 or Chit5-PEG solutions meets with an anti-solvent in an extruder camera with the formation of poly- and interpolyelectrolyte complexes. 

Table 1 presents the data on the hydrodynamic size and zeta potential of the particles formed by Chit5 and Chit5-PEG gelation compared with the situation where alginic acid is used as a counterion. Chit5 in a dilute solution at pH 7.4 forms bulky aggregates because the charge of amino groups is lost in a neutral medium, and the polymer chains stick together (almost zero zeta potential). At the same time, the PEGylated Chit5 has an increased solubility and forms 450 nm nanoparticles. After extrusion, homogeneous neat particles with a hydrodynamic radius of 170 nm (Chit5) and 400 nm (Chit5-PEG) are obtained. In the case of the PEG-chitosan, the hydrodynamic volume is greater due to the PEG hydration shell [45].

Alginic acid, which is a negatively charged polysaccharide, forms electrostatic complexes with chitosan during extrusion; the counterflow of chitosan particles undergoes coacervation due to the compensation of charges of the counterions of alginic acid polyanions. Chit5-PEG (alginic acid) particles have a larger size than Chit5 particles. On the contrary, Chit5-PEG (alginic acid) particles are compacted by polyanion as a crosslinking agent (Table 1).

The recharge of chitosan particles upon coacervation with alginic acid is observed due to the formation of polyelectrolyte complexes (Table 1). This recharge is more pronounced in Chit5 than in Chit5-PEG (in which chitosan chains are partially shielded), which is reflected in the values of the zeta potentials −15 mV vs. −5.5 mV.

### 3.2. Grafting of Polymers with Pyrene

#### 3.2.1. FTIR Spectroscopy

For studying gel or particle formation, the fluorescent approach was applied with the pyrene probe covalently attached to the chitosan chain (Figure 2a). The introduction of a pyrene label into polymers significantly affects the FTIR spectra (Figure 2b); characteristic peaks of C-C (aromatic) oscillations appear (1500–1600 cm^−1^), and the peak intensity of the oscillations of the C=O and N-H in the amide bond increases. On average, 0.5 and 0.4 pyrene molecules were introduced per Chit5 and Chit5-PEG polymer molecule, respectively (based on UV spectroscopy data on pyrene absorption (Figure 2c) and data on the number of amino groups via TNBS titration).

#### 3.2.2. UV and Fluorescence Spectroscopy

The covalent crosslinking of activated pyrene with Chit5 and Chit5-PEG leads to a change in the microenvironment of the fluorophore, which is reflected in the shift of the absorption maxima from 354 to 342–343 nm and from 336 to 327–328 nm (Figure 2c); this spectral change itself can serve as an analytically significant signal (modification or gelation). In the literature, the data on the non-covalent pyrene label is described [46,47,48]; however, it is less sensitive and informative in comparison with the covalent [36,41,49,50]. In the covalent conjugate, the pyrene fluorescence is much brighter than in the free fluorophore (Figure 2d). The kinetic curves of pyrene grafting to polymers is presented in Figure 2d, where the following two stages can be distinguished: (1) 0–2 h—mainly covalent crosslinking of pyrene with chitosan, and (2) 2–72 h—mainly folding of the polymer into a globule due to the use of medium-gel-forming concentrations.

### 3.3. Gel or Nanoparticle Formation

#### 3.3.1. Using a Pyrene Probe to Optimize the Conditions for the Formation of Macrogels (in Volume Phase)

The formation of macrogels is studied in detail using fluorimetry with a pyrene probe (Figure 3). A covalent pyrene is a suitable probe for micelle formation [36,41,46,47,48,49,50,51]; this label is probably also applicable for studying gelation, since the microenvironment in chitosan becomes hydrophobic during gelation. During the formation of a chitosan volume-phase gel, a change in the pyrene microenvironment takes place, and fluorescence quenching is observed.

To monitor the gelation, the following scheme of the experiment was applied: a starting solution where chitosan is soluble at pH 2.5 (all of the amino groups are charged) was used, and gelation was initiated by changing the pH to 3.0, 7.4 or 10.5. Chit5-PEG at pH 3.0 and 7.4 forms particles with a size of 400–500 nm. In an alkaline medium, the solubility of Chit5-PEG decreases, which leads to the formation of sol (visually, but not pieces of polymer as in the case of simple chitosan). At pH 3, the formation of particles practically does not occur for both polymers, but after initiation from pH 8-10, the changes in the pyrene fluorescence are pronounced (Figure 3).

The copolymer Chit5-PEG can self-organize into polymeric micelles in an aqueous solution; hydrophobic Chit5 forms the core of micelles, while hydrophilic PEG chains interact with water molecules in the “PEG-corona” to ensure the solubility of the hydrogel [30,52]. As a result, fluorophore molecules change the fluorescence properties dramatically when the microenvironment changes (when pyrene is incorporated into the hydrophobic core).

#### 3.3.2. Ignition of Dye Fluorescence as Indicator of Volume-Phase Macrogel Formation

Congo red and malachite green were chosen as non-covalent fluorescent probes for monitoring the formation of chitosan particles to study the change in the fluorescence spectrum during the transition from an aqueous solution to a hydrophobic region in a polymer tangle. Figure 4 shows the emission spectra of fluorescence dyes in PBS, DMSO as well as in the composition of polymer chitosan particles. When incorporated into the chitosan particles, there is a sharp increase in the fluorescence of both dyes by about an order of magnitude—this phenomenon can be used as a fluorescent probe for gelation.

#### 3.3.3. pH-Sensitive Volume-Phase Gel Formation

The p*K*_a_ of chitosan ≅ 6.3–6.4; therefore, at a pH below 6, chitosan is soluble, and at neutral or alkaline pH values, it is prone to aggregation due to the loss in the charge of the amino groups. For Chit5-PEG, this effect is leveled by hydrophilic PEG chains that maintain solubility in a wide pH range. Using the probe described above, the pH-dependent character of gelation kinetics was studied. Figure 5 shows the kinetic curves of the fluorescence emission of a fluorescent probe (Congo red) after pH-initiated gel formation. At pH 6.2, Chit5 and Chit5-PEG form nanoparticles (a small increase in fluorescence), while the degree of inclusion of dye in the pegylated chitosan is 2.5 times higher than in the unmodified one (as can be judged from the values of fluorescence emission). At a physiological pH value of 7.4, chitosan aggregates with the formation of microparticles, Chit5-PEG forms nanoparticles (the upward curve) and the capture of Congo red is observed in both cases (Figure 5, Table 1). Chit5-PEG almost instantly formed nano- and microparticles at pH 8.4, and at pH 10.5, the precipitates form large aggregates (the downward curve), and the capture of the dye from the solution is observed. Chit5 aggregates rapidly at pH > 8 and precipitates as large polymer particles. Thus, pegylated chitosan can be used in medicine since it forms stable particles in a wide pH range.

#### 3.3.4. Concentration Dependences of Gelation

Fluorescence polarization characterizes the rotation of plane-polarized light, which is chaotic for small fluorophores, and has specific values for fluorophores that are associated with large molecules (proteins, polymers, DNA, etc.). Figure 6a shows the polarization spectra of Congo red fluorescence depending on the concentration of the added polymer. It is expected that with an increase in the concentration of the polymer, the gel-forming ability increases. It is necessary to find out the critical concentration of the particle formation (depending on the polymer PEGylation) and how gelation is reflected on the fluorescent properties of the probes. Apparently, Chit5 at a concentration of 1 µg/mL and higher forms nano- and microparticles, which is reflected in a sharp increase in the fluorescence polarization. Chit5-PEG forms nanoparticles at a concentration of 1 µg/mL (50–100 nm by DLS, polarization of 0.13), and above 5–10 µg/mL, 400 nm particles (by DLS, polarization of 0.4-0.5) are formed.

In order to study the issue of gelation in more detail, fluorescence emission spectra were recorded depending on the concentration of polymers (Figure 6b). Up to a concentration of 1 mg/mL of Chit5-PEG and 0.1 mg/mL of Chit5, the fluorescence emission of the probe is ignited, which indicates an increase in the dye inclusion degree in the gel particles, accompanied by a shift of the maximum to the long wavelength region by 10–20 nm (Figure 6b,c). With a higher concentration of polymers, fluorescence quenching is observed due to the formation of large aggregates giving a light scattering at 500–550 nm, which is consistent with the fluorescence polarization data.

#### 3.3.5. Temperature Dependences of Gelation

The application of a fluorescent probe to monitor gelation provides valuable information about the flexibility of the polymer chain or hydrophobic site formation, for example, when using a covalent pyrene label [53]. Previously, we described a technique for tracking gelation or micelle formation using rhodamine 6G, but its sensitivity is low and can be improved by using Congo red and malachite green [30,54]. Gelation is accompanied by an increase in the viscosity of the solution and an increase in hydrophobicity due to the aggregation of polymer chains, which can potentially affect the fluorescence of the dye [55,56].

Due to the strengthening of the hydrophobic interactions of chitosan chains at a high temperature, gelation is observed. However, in the case of unmodified chitosan, large aggregates are formed. Contrariwise, pegylated chitosan forms volume-phase gel.

Figure 7 shows the thermograms of Chit5 and Chit5-PEG gelation studied using Congo red probe. The direct and reverse processes of phase transitions were studied, since they are important for understanding the reversibility and stability of the formed particles. When the temperature increases, the fluorescence quenches due to both the temperature factor and gelation. For Chit5, there is almost a coincidence of the forward and reverse path (the process of dye inclusion and particle formation is fast and reversible). For pegylated chitosan, the curve of the reverse process lies significantly below the curve of the direct process and does not come to the starting point, which means that the destruction of the gel from Chit5-PEG is practically not observed. So, once formed, the gel will be stable even at lower temperatures.

#### 3.3.6. Genipin Crosslinked Polymers

Chemical gelation, in contrast to the physical one (described above), can have a number of interesting new properties. The chemical crosslinking of polymer chains can be performed using bifunctional crosslinking agents (diisothiocyanates [57,58], dialdehydes [18,19] or activated derivatives [59,60,61,62,63,64]); however, the issue of safety and biocompatibility is important for their use in medical practice. Genipin is 10,000 times less toxic than glutaraldehyde, and has been successfully used in a number of articles for crosslinking chitosan [16,20,21,65].

When crosslinking polymer chains with genipin, the microenvironment of dyes changes dramatically, but at the same time, its position is quite clearly fixed in the tangle of polymer (Figure 8). The intensity of the fluorescence of the probes increases, and the shift to the wavelength region of the emission maximum due to an increase in the hydrophobicity of the microenvironment also increases. Congo red reacts most vividly to gelation, as in the case of physical gelation. Chemical gelation causes drastic changes in the spectra of fluorescent probes. In this case, the crosslinking of pegylated chitosan leads to the formation of a blue solution of the particles, and insoluble flakes are formed from non-modified chitosan.

### 3.4. Using Förster Resonance Energy Transfer (FRET) as a Tool for Studying Gel Formation and Drug Molecule Encapsulation

FRET is a mechanism of energy transfer between two chromophores (from donor to acceptor), which occurs without intermediate photon emission and is the result of the dipole–dipole interaction between the donor and acceptor [66,67,68,69,70,71,72]. FRET was effectively used to monitor gelation, since the implementation of FRET requires a spatial arrangement of fluorophores less than 10 Å, which can be realized, for example, in a hydrophobic core of micelles or in a tangle of polymers upon gelation.

#### 3.4.1. Tryptophan as a Model of Small (Aromatic) Drug Molecule Encapsulation with FRET Options

##### Volume-Phase Macrogel Formation

We found that upon chitosan copolymer gel formation, a pronounced FRET effect in the pyrene–tryptophan donor–acceptor pare is observed. Figure 9a shows the fluorescence spectra of Chit5-pyrene macrogel in the presence of tryptophan (Trp). The quenching of Trp fluorescence (355 nm) is observed compared to the values in the absence of pyrene. The increase in the proportion of the short-wave pyrene component (at 396 nm) and a decrease in the intensity of the long-wave pyrene component (418 nm) (Figure 9a) is observed with an increase in the Trp concentration due to FRET.

FRET occurs when fluorophores converge, so this phenomenon should be expected during gelation, and this process should be intensified with an increase in the polymer concentration. Figure 9b shows the fluorescence spectra of pyrene-labeled polymers in the presence of Trp at a variable polymer concentration. During gelation, FRET is observed, which is manifested in an increase in pyrene fluorescence while quenching Trp. In the absence of the polymer, FRET practically does not occur (Figure 9b, purple curve), and Trp fluorescence is mainly observed. In the presence of a high concentration of the polymer, FRET is pronounced (Figure 9b, green curve), and pyrene fluorescence is mainly observed; the spectrum is shifted to the long wavelength region, and a wide “tail” fluorescence signal is observed.

##### FRET in Nanogels

Figure 9c shows the fluorescence spectra of nanoparticles based on Chit5 or Chit5-PEG containing pyrene, tryptophan or a mixture. After particle formation (by extrusion), due to pH-initiated coacervation, fluorescence that is several times higher is observed compared to the original pyrene-Chit5 and Chit5-PEG. But in the case of alginic acid used as counterion, the quenching of the pyrene fluorescence is observed. FRET is intensified significantly after the formation of pegylated chitosan particles via extrusion, while for the non-modified chitosan, the spectral differences are smaller.

#### 3.4.2. Ovalbumin as a Model of Large Molecule for Drug Delivery with FRET Options

##### Macrogels

Ovalbumin is a 43 kDa protein, which is the main component of a chicken egg, and it was selected as a biomedically significant (and also Trp-containing) protein model for encapsulation in the chitosan particles. The formation of ovalbumin–chitosan volume-phase gels is accompanied by the intensification of FRET between Tyr, Trp protein and pyrene in the polymer composition (Figure 10a,b). With an increase in the protein concentration to the pyrene-labeled chitosan, the proportion of the (400 nm) pyrene component increases (green, Figure 10a,b). At a chitosan concentration of 0.05 mg/mL (below gel formation concentration), the pyrene peaks strongly overlap with the peak of Trp emission in the protein, and at a concentration of 1 mg/mL (exceeding gel formation concentration), the pyrene peaks are well resolved due to FRET. Thus, gelation is brightly manifested by FRET.

##### Nanogels

An important aspect is the comparison of volume-phase gel and nanogel obtained via extrusion (with 400 nm membrane pores) and after washing of the particles (from soluble components by centrifugation (Figure 10c)). It turns out that Chit5 at a concentration of 0.01 mg/mL does not form particles. At a concentration of 0.1 mg/mL and higher (up to 1 mg/mL), protein–chitosan nanoparticles (with pyrene fluorescence at λmax ≅ 375 nm) are formed. At the same time, Chit5-PEG forms protein–polymer particles at all the concentrations considered (0.01–1 mg/mL), which determines its potential for biomedical use as drug delivery systems or for the creation of gel formulations.

FTIR spectroscopy studies of ovalbumin–chitosan interaction upon gelation.

FTIR provides valuable data on the molecular details of gelation and was used to confirm the fluorescence data. Figure 10d shows the FTIR spectra of ovalbumin in free form and in complex with chitosan particles. In the protein spectrum, the main absorption bands are the following: amide 1 (1600–1700 cm^−1^, ν(C=O)) and amide 2 (1500–1600 cm^−1^, δ(N–H)), ν (C–H) 2580–2970 cm^−1^. In the spectrum of complex particles, a characteristic peak of the oscillations of C–O bonds in Chit5-PEG appears, the shape of the ν(C-H) band changes and the intensity of the absorption bands of amide 1 and amide 2 decreases, which is associated with complexation; we have shown this in an earlier work on concanavalin A [44,73,74,75].

The inclusion of ovalbumin in the particles of pegylated chitosan in gelation is accompanied by pronounced changes in the secondary structure of ovalbumin (Figure 10e). When Chit5-PEG+ovalbumin particles are formed, the intensities of 1638 cm^−1^ and 1657 cm^−1^ components decrease, which means that the proportion of the β-sheet and α-helix decreases by 25% and 10%, respectively, due to protein denaturation and the formation of disordered structures. A peak at 1624 cm^−1^ is increased, which corresponds to the formation of intermolecular aggregates (characteristic for ovalbumin), which was previously described in works on the properties of ovalbumin, adsorbed at the air–water interface [14,76,77]. Thus, ovalbumin is well incorporated in the chitosan particles, which follows from the aggregation of the protein upon chitosan gelation. The property of gelation is promising for the creation of medicinal formulations, dietary supplements and thickeners in the food industry.

## 4. Conclusions

The FRET effect has a growing number of bioanalytical applications. However, little is known about the phenomenon described here, which can be characterized as gel-formation-induced FRET. And we could not find any comprehensive explanation for this in the literature. Indeed, why would a pair of compounds (donor and acceptor) that are not covalently bound, stick together in the result of gel formation, yielding the FRET effect? Intuitively, one might expect that gel formation would lead to the immobilization of Trp—wherever it is localized at that moment. But that is not the case—Trp stacks together with Pyr as soon as a gel is formed, i.e., the presence of a gel forces a stacking interaction to happen, which is not possible in the solution due to diffusion. Obviously, gel formation does not only mean a higher viscosity, but it is also a quasi-regular structure, promoting quasi-regular interactions of all components involved, including Trp and Pyr. We believe the mechanism of this process deserves a fundamental investigation. 

But, regardless, we demonstrate here that the FRET effect can be used in a new kind of practical application, as a probe with unmatched sensitivity to monitor the nanogel formation process. In this context, the FRET effect yields valuable real-time information and appears to be much more specific than the existing alternatives. The study of a control system with a similar polymer composition that forms macrogels, which allows us to visually (for example Figure 6) or turbidimetrically detect gel formation, shows that the FRET effect is observed symbiotically with the observed process of gel formation. This result indicates the specificity of the FRET effect to the formation of the gel (which is also translated to the case of nanogels). This phenomenon may help in the development of nanogels with a wide range of potential applications, including food and biomedical technologies and research.

## Figures and Tables

**Figure 1 jfb-14-00401-f001:**
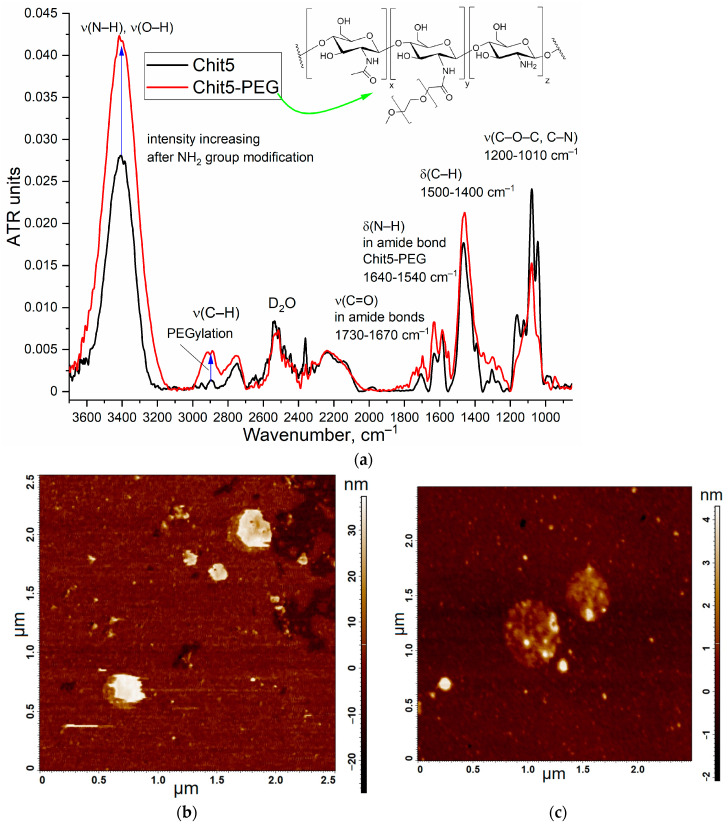
(**a**) FTIR spectra of Chit5 and Chit5-PEG in D_2_O. Atomic force microscopy images of particles formed from (**b**) Chit5 and (**c**) Chit5-PEG. T = 22 °C.

**Figure 2 jfb-14-00401-f002:**
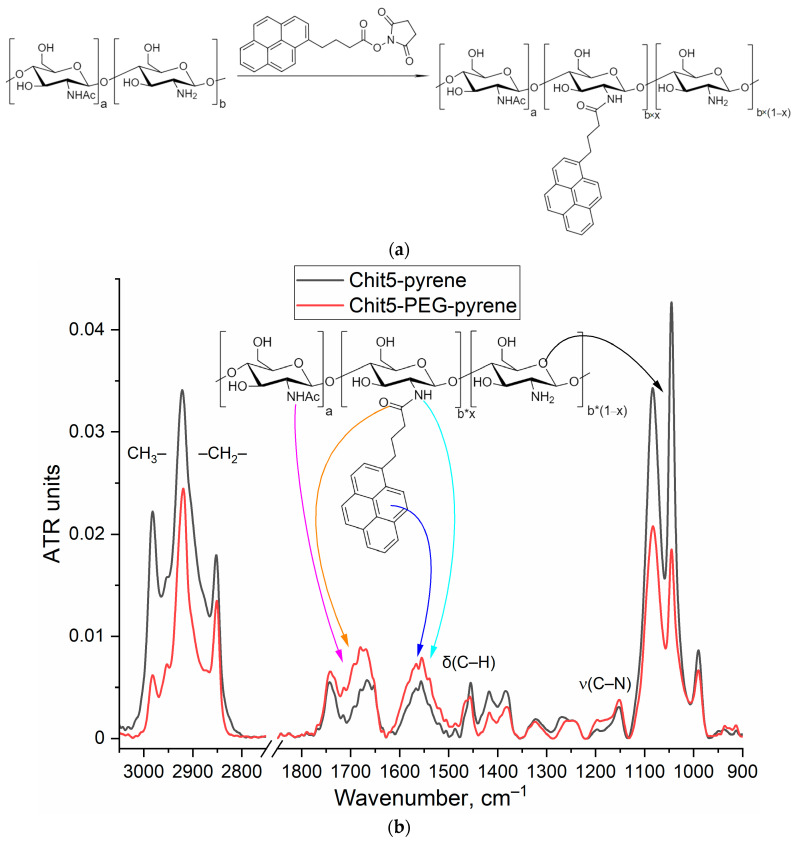
(**a**) Scheme of synthesis of pyrene-grafted Chit5 (for Chit5-PEG, the reaction is similar). (**b**) FTIR spectra of Chit5-pyrene and Chit5-PEG-pyrene. PBS (0.01 M, pH 7.4). T = 22 °C. (**c**) UV spectra of activated pyrene and pyrene crosslinked with Chit5, Chit5-PEG in PBS-DMSO (0.01% DMSO *v*/*v*). (**d**) Fluorescence spectra of pyrene during conjugation with polymers and corresponding kinetic curves. PBS (0.01 M, pH 7.4). T = 40 °C.

**Figure 3 jfb-14-00401-f003:**
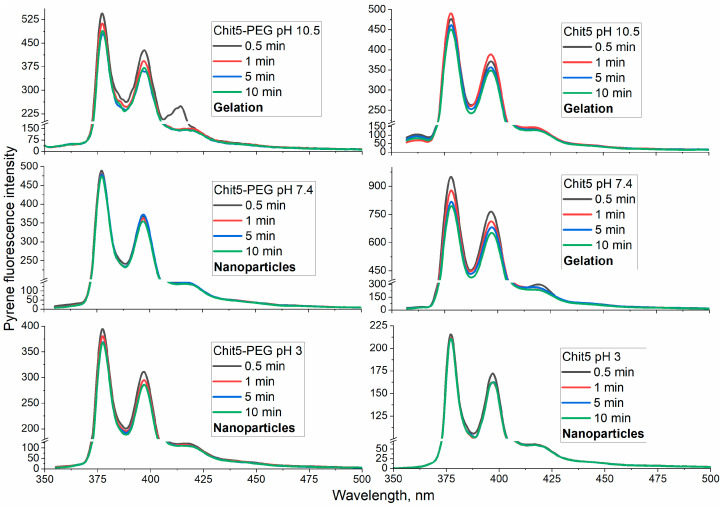
Fluorescence spectra of pyrene-grafted Chit5 and Chit5-PEG during gel or nanoparticle formation at different pH levels. C_polymer_ = 1 mg/mL. Gelation was pH induced. T = 40 °C. λ_exci_ (pyrene) = 330 nm.

**Figure 4 jfb-14-00401-f004:**
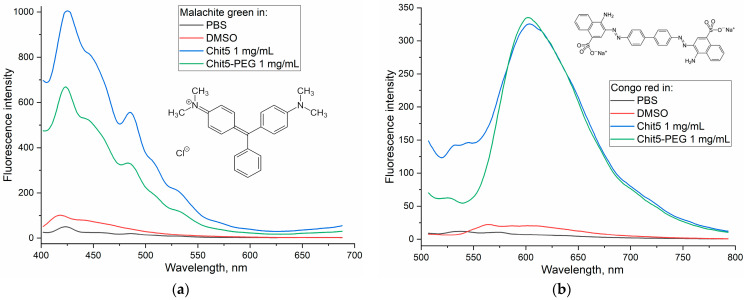
Fluorescence spectra of (**a**) Congo red and (**b**) malachite green (10 μg/mL). T = 40 °C. λ_exci_ (malachite green) = 370 nm, λ_exci_ (Congo red) = 480 nm.

**Figure 5 jfb-14-00401-f005:**
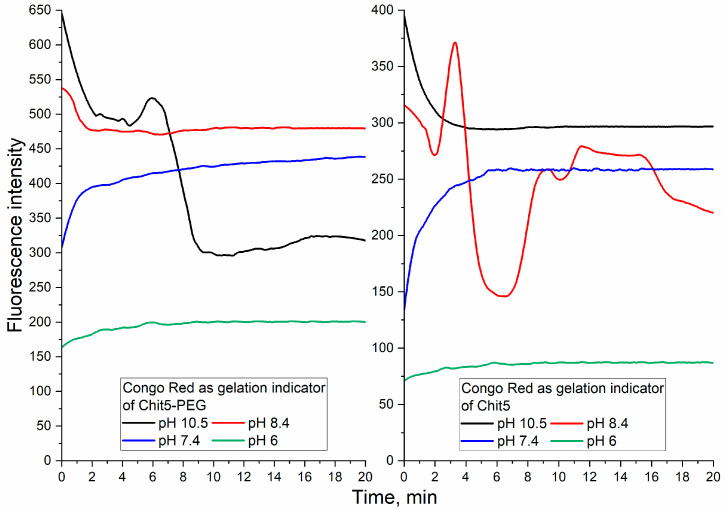
Kinetic curves of Congo red (10 μg/mL) fluorescence emission during particle and gel formation by Chit5 and Chit5-PEG. T = 40 °C. λ_exci_ (Congo red) = 480 nm. The curves were normalized to the values of free dye emission at a given pH.

**Figure 6 jfb-14-00401-f006:**
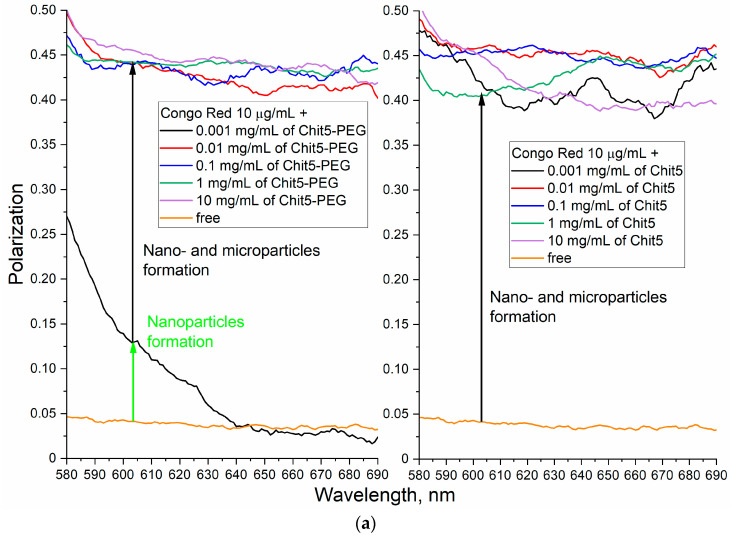
Fluorescence approach to study the concentration dependence of gelation: (**a**) fluorescence polarization spectra, (**b**) emission spectra, (**c**) corresponding dependences of maxima positions and intensity. Congo red (10 μg/mL) was used as probe. PBS (pH 7.4). T = 40 °C. λ_exci_ (Congo red) = 480 nm. (**d**,**e**) Photos of samples after 30 min of incubation. Chit5 forms colored polymer pieces, and Chit5-PEG forms a stable colloidal system.

**Figure 7 jfb-14-00401-f007:**
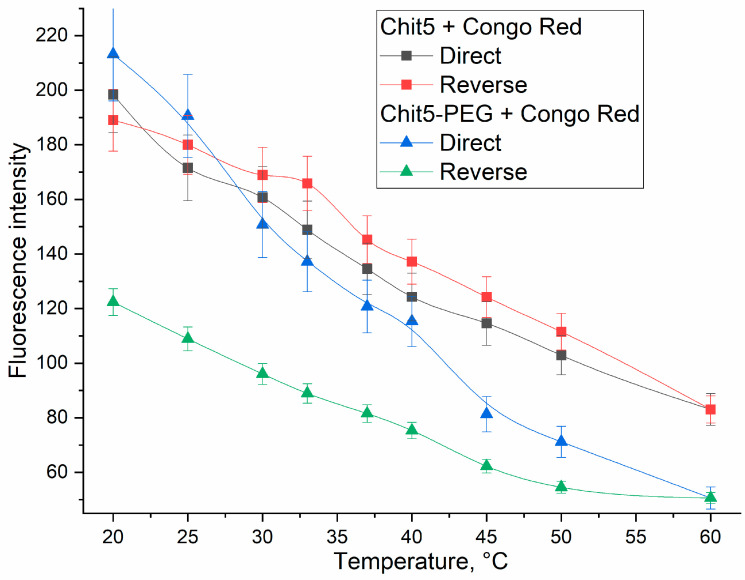
Thermo curves of Congo red (10 μg/mL) fluorescence emission during particle and gel formation by Chit5 and Chit5-PEG. λ_exci_ (Congo red) = 480 nm. PBS (pH 7.4).

**Figure 8 jfb-14-00401-f008:**
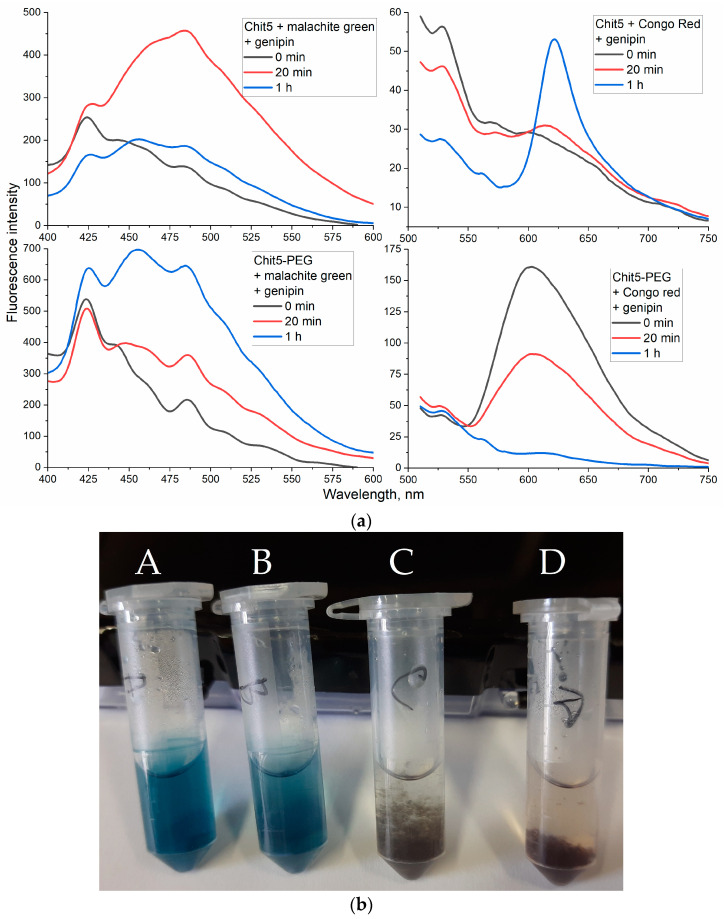
(**a**) Fluorescence spectra of Congo red and malachite green (10 μg/mL) during Chit5 and Chit5-PEG chains’ genipin (1 mg/mL) crosslinking. T (reaction), 70 °C; T (spectra registration), 40 °C. λ_exci_ (malachite green) = 370 nm, λ_exci_ (Congo red) = 480 nm. PBS (pH 7.4). (**b**) Photos of samples after 1 h of reaction; A—malachite green in Chit5-PEG5, B—Congo red in Chit5-PEG, C—malachite green in Chit5, D—Congo red in Chit5.

**Figure 9 jfb-14-00401-f009:**
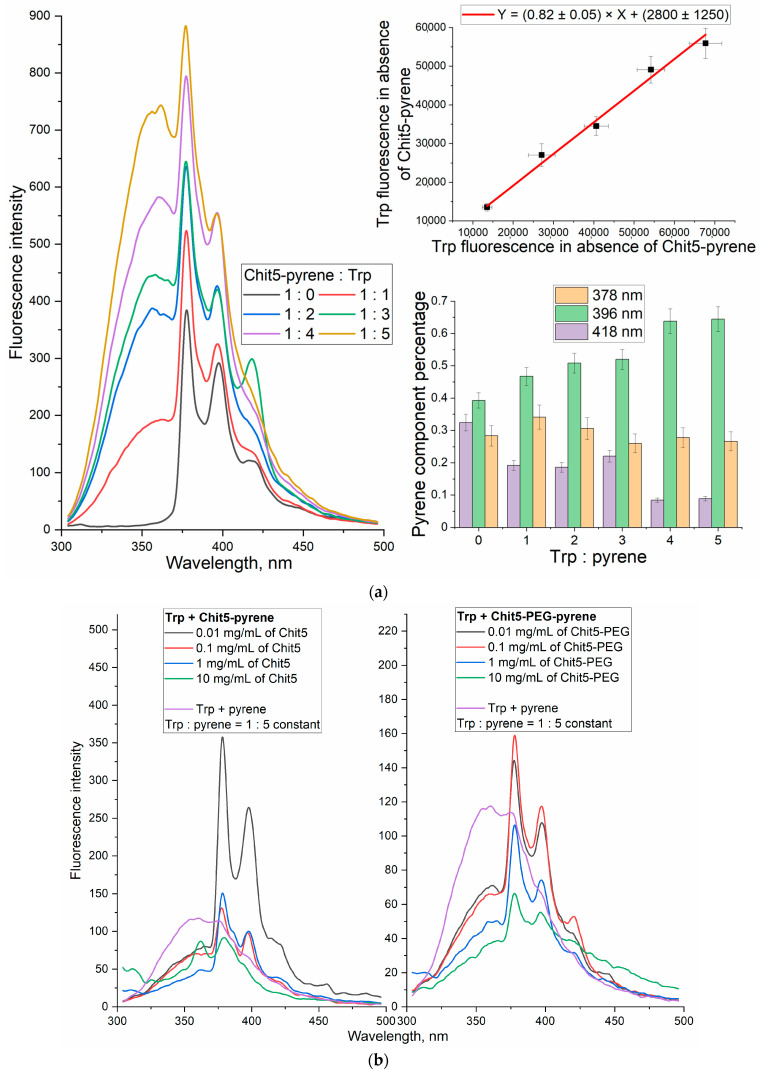
(**a**) The fluorescence spectra of Chit5-pyrene in the absence and in the presence of different concentrations of tryptophan, and the corresponding dependences of integral intensities and percentages of fluorescence. (**b**) Fluorescence spectra of Chit5-pyrene, Chit5-PEG-pyrene, pyrene in the presence of tryptophan, depending on the concentration of the polymer. PBS (pH 7.4). (**c**) Fluorescence spectra of the following particles: Chit5, Chit5-PEG with pyrene (covalent), Trp or its mixture—native, 400 nm extrusion in solution with pH 10; 400 nm extrusion in solution with alginic acid. T = 40 °C.

**Figure 10 jfb-14-00401-f010:**
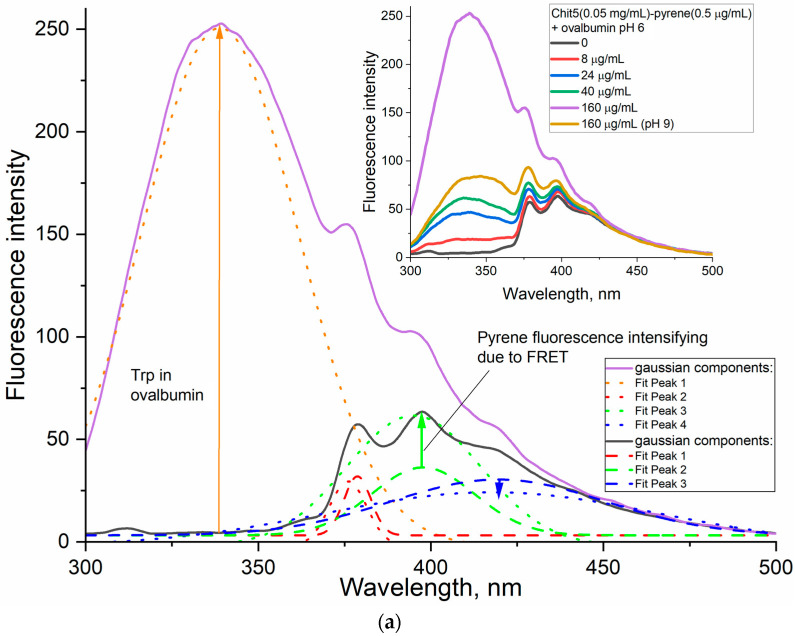
(**a**) The fluorescence spectra of Chit5-pyrene (0.05 mg/mL of Chit5, 0.5 μg/mL of pyrene) in the absence and in the presence of different concentrations of ovalbumin, and deconvolution of peaks with gaussians. (**b**) The fluorescence spectra of Chit5-pyrene (1 mg/mL of Chit5, 0.5 μg/mL of pyrene) in the absence and in the presence of different concentrations of ovalbumin, and deconvolution of peaks with gaussians. Phosphate buffer (10 mM, pH 6). Borate buffer (10 mM, pH 9). (**c**) The fluorescence spectra of particles formed by Chit5, Chit5-PEG alone (black) and after extrusion in borate buffer with pH 7.4 containing ovalbumin (red) or with subsequent centrifuge purification and separation of nanoparticles (blue). (**d**) FTIR spectra of free ovalbumin and ovalbumin (Chit5-PEG) particles. (**e**) FTIR spectra deconvolution on the following components: 1638 cm^−1^ β-sheets, 1657 cm^−1^ α-helix, 1686 and 1676 cm^−1^ β -turns, 1626—intermolecular β-sheets (aggregates). T = 40 °C.

**Table 1 jfb-14-00401-t001:** Physicochemical parameters of particles formed by Chit5 and Chit5-PEG compacted with alginic acid (30–50 kDa, 2-fold molar excess). Concentration of polymers is 0.1 mg/mL. PBS (0.01 M, pH 7.4). Extrusion through a 400 nm membrane was carried out using a syringe for injecting a solution of Chit5 or Chit5-PEG and a receiver syringe containing a borate buffer solution with pH 10 or an alginic acid solution (pH 7.4) to initiate particle formation.

Polymer	Sample	Hydrodynamic Diameter of Particles, nm	ζ Potential, mV
Chit5 (5 kDa, 90% degree of deacetylation)	Native	>1 μm	0.7 ± 0.3
	After extrusion in pH 10	170 ± 40	−1.9 ± 0.8
	After extrusion in alginic acid solution	260 ± 40	−15 ± 2
Chit5-PEG (1/1 *w*/*w*, 10 kDa) *	Native	450 ± 70	3.5 ± 0.9
	After extrusion in pH 10	400 ± 30	−0.3 ± 0.2
	After extrusion in alginic acid solution	190 ± 10	−5.5 ± 1.9

* Determined via FTIR (Figure 1) and by the number of amino groups in Chit5 using TNBS titration before and after modification.

## Data Availability

The data presented in this study are available in the main text.

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
