# Peer review of "Fluorescent Probes with Förster Resonance Energy Transfer Function for Monitoring the Gelation and Formation of Nanoparticles Based on Chitosan Copolymers"

_jfb, 2023, doi:10.3390/jfb14080401_

Round 1

Reviewer 1 Report

This manuscript can be accepted after minor revision. My comments are as follows:

Some sentences need to be revised.  They can cause confusion. I list some examples: 1)The 

inclusion of ovalbumin in polymer particles is interesting in aspects of potential applica- 

tion in the biomedicine (including vaccine formulations), food industry for the creation of 

foams and emulsions, gels, mousses, in cosmetology. 2) Upon nanogel formation FRET effect is observed manifested in sharp 24

increase if the fluorescence intensity of the pyrene (acceptor). 

The authors talk about particles with the size of 170 or 400 nm. Why I did not see any image of TEM ? 

The authors should show some representative images. 

Ref.66 did not show the journal name ? 

What is the meaning of " 400 nm extrusion in solution with alginic acid"? The authors may have to organize the text in the figures.

Minor revision on the writing / English Language is required.

Reviewer 2 Report

The authors tried to present the synthesis of chitosan based nanoparticles and monitored their gelation and formation using fluorescent probes with FRET function. Manuscript is well organized and presents new findings in the field. However, some important points need to be carefully addressed before possible publication at this prestigious journal.

1.      English language and scientific style of the manuscript should be thoroughly improved as in current state the manuscript is very difficult to understand for the readers.  Better to use a professional English editing service. A few points are as follows:

First sentence of the abstract should be divided into two parts.

Abstract reads: “The paper presents three based fluorescent probes…” what does it mean? Need to be corrected.

Abstract reads: “particles formation accompanied by sharply increases in the fluorescence…” grammatically incorrect.

Again “A sensitive approach for tracking gelation and including of small drug molecules…” grammatically incorrect.

Again, same issue: “Thus, the application of the fluorescent probes with FRET function for monitoring the gelation and nanoparticles formation as well as drug encapsulation in chitosan-based parts, which have broad prospects in biomedical practice and food industry.” Meaningless sentence.

I am sorry that I am unable to mention more English modifications. Can you please rewrite the whole manuscript or get it edited from an expert native speaker or professional editing service?

2.      What is the novelty of your work? Just presenting some data is not a scientific paper.

Can the formation of other types of nanoparticles with polymers be monitored using the current approach? Cite and explain this work as an example: Templated synthesis of crystalline mesoporous CeO2 with organosilane-containing polymers: balancing porosity, crystallinity and catalytic activity[J]. Materials Futures 2022, 1(2): 025302. doi: 10.1088/2752-5724/ac7605

3.      You mentioned that the intensity of FTIR peak at 3400 cm-1 increases after PEG modification. However, same results can be obtained by changing the concentration of the sample.

4.      Moreover, OH peak overlaps with NH peak, how do you assign it to NH?

5.      Some FTIR peaks disappear after PEG functionalization, such as in the range of 1400-1000 cm-1. Why is that? What does it indicate?

6.      FTIR pattern of Chit5-pyrene is better with stronger intensity than that of Chit5-PEG pyrene. Why is it so? Is it in contact with the results of Fig. 1?

7.      UV spectra shows shift in pyrene peaks after functionalization with chitosan. Does it indicate any change in its structure? How can you confirm the structure preservation?

8.      You mentioned that change in pH helps increase the intensity difference between samples prepared at different time durations. However, there is no obvious difference after 5 min at any pH. Why? 

9.      In Fig. 5 chit5 shows better fluorescence in case of both the congo red and malachite green. Same was the case for pyrene modification. Why you modify chit5 with PEG, while its deteriorating the properties of chit5.

10.  The study on Förster resonance energy transfer as a tool for gel formation need to be clarified. Signify the main findings and mechanism.

11.  It was shown that particles load both small drug and large drug molecules. How can you make sure these drug molecules will stay loaded during drug delivery to the body?

12.  Can the FRET help notify the delivery of drug to a specific point in the body?

13.  Since the chitosan and its derivatives are biocompatible and find potential applications in medical fields, can these be combined with other sensors such as these: (1. Materials Futures 2022, 1(2): 022401. doi: 10.1088/2752-5724/ac48a3. 2. Materials Futures 2022, 1(2): 023502. doi: 10.1088/2752-5724/ac7068). Please cite and explain.

14.  Too much data is presented in the manuscript. I think only present the main findings in main text body and include others in supporting information.

Minor editing of English language required

Reviewer 3 Report

The authors aimed, in an approach based on the fluorescent methods (with the FRET option), to study the physicochemical aspects of particle formation depending on pH, temperature, and the presence of a crosslinking agent, which is promising for creating optimal biomedical gels, ointments, and dosage formulations. In fact, the inclusion of ovalbumin in polymer particles is interesting in aspects of potential application in the biomedicine (including vaccine formulations), food industry for the creation of foams and emulsions, gels, mousses, in cosmetology.

The study covers some issues that have been overlooked in other similar topics. The structure of the manuscript appears adequate and well divided in the sections. Moreover, the study is easy to follow, but some issues should be improved. Some of the comments that would improve the overall quality of the study are:

I-) Authors must pay attention to the technical terms acronyms they used in the text

II-) Conclusion Section: This paragraph required a general revision to eliminate redundant sentences and to add some "take-home message".
